# Bearing Fault Diagnosis Using Piecewise Aggregate Approximation and Complete Ensemble Empirical Mode Decomposition with Adaptive Noise

**DOI:** 10.3390/s22176599

**Published:** 2022-09-01

**Authors:** Lei Hu, Ligui Wang, Yanlu Chen, Niaoqing Hu, Yu Jiang

**Affiliations:** 1College of Railway Transportation, Hunan University of Technology, Zhuzhou 412007, China; 2Hunan Provincial Key Laboratory of Health Maintenance for Mechanical Equipment, Hunan University of Science and Technology, Xiangtan 411201, China; 3Laboratory of Science and Technology on Integrated Logistics Support, National University of Defense Technology, Changsha 410073, China

**Keywords:** rolling bearings, fault diagnosis, piecewise aggregate approximation, CEEMDAN

## Abstract

Complete ensemble empirical mode decomposition with adaptive noise (CEEMDAN) effectively separates the fault vibration signals of rolling bearings and improves the diagnosis of rolling bearing faults. However, CEEMDAN has high memory requirements and low computational efficiency. In each iteration of CEEMDAN, fault vibration signals are added with noises, both the vibration signals added with noises and the added noises are decomposed with classical empirical mode decomposition (EMD). This paper proposes a rolling bearing fault diagnosis method that combines piecewise aggregate approximation (PAA) with CEEMDAN. PAA enables CEEMDAN to decompose long signals and to achieve enhanced diagnosis. In particular, the method first yields the vibration envelope using bandpass filtering and demodulation, then compresses the envelope using PAA, and finally decomposes the compressed signal with CEEMDAN. Test data verification results show that the proposed method is more effective and more efficient than CEEMDAN.

## 1. Introduction

Rolling bearings are one of the most widely used components in rotating machinery. Failure of rolling bearings are one of the most frequent reasons for machine breakdown. Thus, fault diagnosis of rolling bearings is crucial to ensure the operational efficiency and reliability of engineering systems [1,2]. When a fault bearing rotates, a localised defect on the outer or inner race is struck by the rollers, or a localised defect on a roller strikes the inner and outer races. High-frequency resonances are excited and presented as impact transients. The periodicity of the successive impact transients is expressed as characteristic fault frequencies [2]. The vibration of fault bearing is recognised as the modulation between the components of low fault frequency $f_{F}$ and high natural frequency $f_{n}$, as shown in Figure 1. It is the most classic bearing fault diagnosis method to obtain the envelope spectrum or squared envelope spectrum using bandpass filtering and demodulation [3]. Finding the optimal frequency band for filtering is critical for the envelope analysis [4]. Some successful tools, such as fast Kurtogram [5], the improved Kurtogram based on wavelet packet transform [6], protrugram [7], and Autogram [8], have been developed for finding the optimal band.

Empirical mode decomposition (EMD) is another widely used method for bearing fault diagnosis. EMD decomposes a signal into a set of intrinsic mode functions (IMFs) and a residue signal [9]. The IMFs are narrow-band components and indicate the natural oscillatory mode imbedded in the original signal [10]. As EMD is effective for nonlinear, non-stationary signals with both Gaussian and non-Gaussian noise, it has been applied with success in different fields, including bearing fault diagnosis [11], planetary gearbox fault diagnosis [12], railway structural wavelength identification [13], automatic sleep scoring [14], etc. However, EMD suffers from endpoint effects and mode mixing. As for mode mixing, a single IMF consists of signals of widely disparate scales, or a signal of a similar scale resides in different IMF components [15]. The mode mixing is the major drawback of the EMD. Ensemble EMD (EEMD) is developed to suppress mode mixing by adding assisted noises to improve the extrema distribution of the signal [16]. However, the IMFs generated by EEMD contain residual noise, and different numbers of IMFs can be generated as different assisted noises are added to the signal to be decomposed. In order to solve the problem that the IMFs are contaminated by residue noise, complementary EEMD (CEEMD) is presented via adding noises in pairs with opposite signs to the targeted signal [17,18]. However, the completeness property is not proven, and different noisy copies of the signal can produce a different number of modes. How to choose proper parameters is also a problem for CEEMD. A further improved algorithm named CEEMD with adaptive noise (CEEMDAN) is proposed to solve the problem of incomplete decomposition by adding particular noise to the signal, which in turn reduces the residual noise in the IMFs [19]. CEEMDAN has been applied in the fields of biomedical engineering [20], energy economics [21], and fault diagnosis [22,23]. In each iterative layer of CEEMDAN, *N* signals added with noises, as well as the *N* assisted noises (*N* is the number of overall averages), are decomposed. Thus, CEEMDAN takes up a lot of memory, and is of low computational efficiency especially for long signal analysing.

A longer signal brings more robust information. For a signal of $x\left(t\right)$, its Fourier transform is:(1)$$F\left(\omega \right)=F\left[f\left(t\right)\right]=\int_{t_{1}}^{t_{2}}f\left(t\right)e^{-j\omega t}dt.$$

It can be seen from Equation (1) that a frequency component reflects the average energy of the periodic component over the entire test period of $t_{2}-t_{1}$. Theoretically, the longer the signal is, the more times that a component is averaged, and the clearer the spectrum will be. Figure 2a shows the simulation signal of a bearing with background noise. The signal length is L=100 s and the signal noise ratio (SNR) is −18.091 dB. The fault characteristic frequency is 15 Hz. Signals with a length of 2 s, 10 s, 30 s, and 100 s are selected for envelope analysis. The corresponding envelope spectra yielded are shown in Figure 2b–e, respectively. It can be seen that the harmonics of fault frequency, which cannot be seen in the spectrum of 2 s, can be seen in the spectra of 10 s, 30 s, and 100 s. Although the harmonics of 10 s, 30 s, and 100 s have nearly equal amplitudes, the harmonics become increasingly clearer from Figure 2c–e, as the longer the signal length is, the better the background noise is reduced.

However, longer signals of high sampling frequency also increase the requirement of computing hardware, which can be a challenge especially for the application cases of edge computing. Particularly, as the natural frequency $f_{n}$ is as high as thousands (or even tens of thousands) of Hz, the sampling frequency of bearing vibration, $f_{S}$, is set to be tens of thousands of Hz according to the Nyquist sampling theorem. Thus, it is natural to compress the signal before processing it using algorithms of high complexity. The technique of compressed sensing achieves data acquisition and compression at the same time. The measurements that compressed sensing obtains are nonadaptive linear projections of the original signals. And the original signals can be reconstructed with the measurements using recovery algorithms [24]. Compressed sensing is originally used for image processing in the fields of medical imaging [25,26,27], radar imaging [28,29], astronomy [30,31], face recognition [32,33], etc. Compressed sensing is also introduced for machinery fault diagnosis to obtain sparse representation of original signals and to extract fault features from the compressed signals [34,35,36]. The major drawback of compressed sensing for fault diagnosis is that the compression is not supervised with prior knowledge. Some classical diagnosis methods, such as envelope analysis and EMD, are not applicable any more for the compressed signals. In addition, loss of fault information is inevitable when reconstructing the original signals from the compressed signals.

Piecewise aggregate approximation (PAA) is a far easier method that can be used for signal compression [37,38]. An improved PAA is proposed to take fluctuating trends into account as well [39]. PAA first divides the time series into *N* segments equally and uses the average of each segment as an approximate representation of that segment. In this process, the original time series with *L* samples is compressed into a signal of *N* samples, which can be regarded as a process of dimensionality reduction. The equivalent sampling frequency of the compressed signal is $f_{\mathrm{ES}}=f_{S}\times N/L$, where $f_{S}$ is the sampling frequency of the original signal. Thus, there is information loss for components whose frequencies are larger than $f_{\mathrm{ES}}/2.56$. 

In order to obtain reliable diagnostic results using long signals, a method combining PAA and CEEMDAN is proposed. In order to overcome the problem that CEEMDAN has large memory requirements and low computational efficiency, PAA is introduced to compress the signals before decomposing them. Moreover, in order to avoid information loss caused by signal compression, the traditional envelope analysis method is applied and PAA is performed on the envelopes instead of the original signals. Validations are carried out with signals collected from real rolling bearings.

## 2. Methodology

### 2.1. Complete Ensemble Empirical Mode Decomposition with Adaptive Noise (CEEMDAN)

CEEMDAN is an improved algorithm of EMD and EEMD, which overcomes the shortcomings of EEMD, as mentioned in Section 1. The flow chart of CEEMDAN is shown in Figure 3.

Assuming y is the signal to be decomposed, CEEMDAN is performed to decompose the signal y, and the IMF obtained by layer i decomposition is expressed as $\bar{C_{i}}$, i=1, 2, ⋯, I, where I is the number of layers of decomposition, and the decomposition steps are as follows: 

(1)First layer decomposition, i.e., i=1.

① Adding white noise $v^{j}$ to the signal of y yields a new signal of $y+\varepsilon_{i=1}v^{j}$, where j=1,2,…,N, N is the number of adding white noise, and $\varepsilon_{i=1}$ is the amplitude of white noise.

② Decomposing the new signal of $y+\varepsilon_{i=1}v^{j}$ with EMD yields a series of IMFs, and the first IMF is presented as$\;E_{1\mathrm{st},i=1}^{j}$.

③ Ensemble averaging of *N* IMFs $E_{1\mathrm{st},i=1}^{j}$ yields the ith $\left(i=1\right)$ IMF of CEEMDAN:(2)$$\bar{C_{i=1}}=\frac{1}{N}\sum_{j=1}^{N}E_{1\mathrm{st},i=1}^{j}$$

④ Removing the first IMF of $\bar{C_{i=1}}$ from y yields the residual of $r_{i=1}$:(3)$$r_{i=1}=y-\bar{C_{i=1}}$$

(2)Second layer decomposition, i=2.

① Decomposing $v^{j}$ with EMD yields a series of IMFs, the first of which is presented as $E_{1}\left(v^{j}\right)$. Adding $E_{1}\left(v^{j}\right)$ as noise to the residual $r_{i-1}$ yields a new signal of $r_{i-1}+\varepsilon_{i}E_{1}\left(v^{j}\right)$.

② Decomposing the new signal $r_{i-1}+\varepsilon_{i}E_{1}\left(v^{j}\right)$ with EMD yields a series of IMFs, the first of which is presented as $E_{1\mathrm{st},i}^{j}$.

③ Ensemble averaging of *N* IMFs $E_{1\mathrm{st},i}^{j}$ yields the ith IMF of CEEMDAN:(4)$$\bar{C_{i}}=\frac{1}{N}\sum_{j=1}^{N}E_{1\mathrm{st},i}^{j}$$

④ Removing the ith IMF of $\bar{C_{i}}$ from $r_{i-1}$ yields the residual of $r_{i}$:(5)$$r_{i}=r_{i-1}-\bar{C_{i}}$$

(3)The above steps are repeated until the residual signal obtained is a monotone function and cannot be further decomposed, at which point the algorithm ends. At last, the signal to be decomposed is presented as:

(6)$$y=\sum_{i=1}^{I}\bar{C_{i}}+r_{I}$$
in which I is the number of IMFs and $r_{I}$ is last residual signal.

### 2.2. Piecewise Aggregate Approximation

It can be seen that CEEMDAN has large memory requirements and low computational efficiency, as in each iteration of CEEMDAN, tens of fault vibration signals added with assisted noises, as well as the assisted noises, are decomposed with classical EMD. To solve the problem of low computational efficiency, PAA is introduced to compress the signals before performing CEEMDAN.

PAA compresses a large amount of time series data while keeping as many original features of the data as possible. Assuming that the test signal is $x=\left\{x_{i}\right\}$, the sampling frequency is $f_{S}$, and the signal length is *L*. PAA defines a constant window w, then divides the sample sequence x into N equal segments, N=⎣L/w⎦, and finally calculates the mean of each segment:(7)$$p_{n}=\frac{1}{w}\sum_{i=w(n-1)+1}^{nw}x_{i}\;,n=1,\;2,\;\cdots ,\;N$$

The new sequence $p=\left(p_{1},p_{2,}\ldots ,p_{N}\right)$ is the obtained compressed signal. It can be seen that the equivalent sampling frequency of the compressed signal is $f_{\mathrm{ES}}=f_{S}/w$. The larger w is, the smaller the samples that the compressed signal has obtained.

### 2.3. Diagnosis Flowchart

Figure 4 shows the flow chart of the proposed method, which consists of five main steps: optimal band selection for filtering, bandpass filtering and demodulation, PAA, CEEMDAN, and spectra analysis. The steps are depicted as follows:

(1)Optimal filtering band selection.

In order to enhance the modulation signal of low fault frequency and high natural frequency, finding an optimal resonance band for bandpass filtering is critical. The fast Kurtogram, which finds the optimal band according to the kurtosis of the filtered time signal in different filter banks, has been proven to be a practical tool in bearing fault diagnosis. Thus, the fast Kurtogram is introduced for optimal filtering band selection.

(2)Bandpass filtering and demodulation.

Bandpass filtering enhances the modulation signal of low fault frequency and high natural frequency, while demodulation obtains the envelope signal y of low fault frequency, $y=\left|x+iH\left(x\right)\right|$, where x is the filtered signal and $H\left(x\right)$ is the Hilbert transform of x. The envelope consists of components of low frequencies, including the harmonics of fault frequencies. As the fault frequencies are far smaller than the natural frequency, the envelope can be compressed to obtain a signal whose equivalent sampling frequency is far smaller than the original sampling frequency.

(3)Signal compression.

PAA is introduced to compress the envelope yielded in the second step. PAA first divides the envelope into N segments of equal length w, N=⎣L/w⎦, where L is the length of the envelope. Then, PAA uses the mean $p_{i}$ of each segment as an approximate representation of the segment. The obtained compressed signal is $p=\left\{p_{i}\right\}$.

The window size, or the segment length, w, is the only unknown parameter of PAA. In addition, w can be set according to the requirement for the equivalent sampling frequency $f_{\mathrm{ES}}$ of the compressed signal. As for the envelope of the bearing fault signal, the interesting components are the harmonics of bearing fault characteristic frequencies, which include the ball pass frequency of outer race $f_{\mathrm{BPFO}}$, the ball pass frequency of inner race $f_{\mathrm{BPFI}}$, the ball spin frequency $f_{\mathrm{BS}}$, and the cage frequency $f_{C}$. The maximum of the bearing fault characteristic frequencies, $f_{\max}=\left(f_{\mathrm{BPFO}},f_{\mathrm{BPFI}},f_{\mathrm{BS}},f_{C}\right)$, is generally $f_{\mathrm{BPFI}}$. According to the Nyquist sampling theorem, the equivalent sampling frequency of the compressed signal should satisfy the condition of $f_{\mathrm{ES}}\geq 2.56Zf_{\max}$, in which Z is the max order of fault frequency harmonics. Therefore, the window size meets the inequality of:(8)$$w\leq f_{S}/\left(2.56Zf_{\max}\right).$$

(4)CEEMDAN.

Following the steps of CEEMDAN described in Section 2.1, the compressed signal is decomposed, and a series of IMFs is obtained.

(5)Spectrum analysis.

Spectrum analysis is performed on the IMFs obtained to find the interesting IMFs whose frequency bands cover the fault characteristic frequency. Fault diagnosis of rolling element bearing is finally achieved according to the spectra of the interesting IMFs.

### 2.4. Remarks

PAA is simple, but the envelope waveform of impact transients is well retained in the compressed signal. The reason is that signal compression is supervised with prior knowledge. Particularly, PAA compresses the envelope instead of the original signal. The series of impact transients produced successively by a localised defect are recognised as the modulation between the low-frequency fault components and high-frequency resonances. Thus, bearing vibration is collected with high sampling frequency, and compressing the original vibration signal causes the information loss of the high-frequency resonance; while the diagnostic information in the demodulated envelope is the low-frequency fault components, and the information will be kept in the compressed signal as long as the equivalent sampling frequency is larger than 2.56 multiples of interesting frequencies.

## 3. Experiment Validation

Bearing fault simulation tests are carried out on the test bench, as shown in Figure 5. The test bench consists of a driving motor, a bearing-supported rotating shaft, an inertia wheel for providing radial load, a belt drive mechanism, a gearbox, a crank connecting rod mechanism, and a reciprocating mechanism. The bearing seeded with defect is mounted in the bearing housing closer to the motor. The seeded defect is a localised crack with both a width and depth of 0.2 mm. The bearing is a deep groove ball contact bearing, the model is MB-ER-10K. The fault characteristic frequencies are $f_{\mathrm{BPFO}}=3.052f_{r}$, $f_{\mathrm{BPFI}}=4.948f_{r}$, $f_{\mathrm{BS}}=1.992f_{r}$, and $f_{C}=0.382f_{r}$, where $f_{r}$ is the shaft frequency. Vibration signals were collected using accelerometers of the PCB Model 608A11, whose bandwidth is of 0.5 Hz~9 kHz. The sampling frequency was set as $f_{S}=25.6$ kHz.

The maximum of the bearing fault characteristic frequencies is $f_{\max}=f_{\mathrm{BPFI}}$. Assuming that Z=5 orders of fault frequency harmonics are supposed to be retained in the compressed signals, it yields the condition of the window length, $w\leq 404.20/f_{r}$, according to Equation (8).

### 3.1. Validation for Outer Race Defect Case

The vibration signal of a bearing with an outer race defect is shown in Figure 6a. The signal length is *L* = 19 s, the shaft speed is $f_{r}=14.1184$ Hz, and the corresponding fault frequency is $f_{\mathrm{BPFO}}=43.0894$ Hz. The proposed method combining CEEMDAN and PAA was used to analyse the signal. Firstly, analysing the signal with fast Kurtogram yields the diagram, as shown in Figure 7. It can be seen that the center frequency of the optimal band is 10,667 Hz, the bandwidth is 4267 Hz, and the corresponding optimal filtering band is 8533.5~12,800.5 Hz. The filtered signal for the optimal filtering band is shown in Figure 6b. The envelope of the filtered signal is shown in Figure 6c.

Performing PAA to compress the envelope yields the result shown in Figure 6d. The window length is set to be w=20 as $w\leq 404.20/f_{r}$ and $f_{r}=14.1184$ Hz. The equivalent sampling frequency of the compressed signal is $f_{\mathrm{ES}}=1.28$ kHz. Partial enlarging the envelope of Figure 6c yields Figure 6e, and partial enlarging the compressed signal of Figure 6d yields Figure 6f. Comparing Figure 6d and Figure 6c, and Figure 6f and Figure 6e, it can be seen that although the compressed signal has smaller amplitudes than the envelope does, they share the same waveform of impulses.

Decomposing the compressed signal with CEEMDAN yields 16 IMFs. The spectra of the IMF 2~IMF 7 are shown in Figure 8, from which the component of $f_{\mathrm{BPFO}}$ and its high order harmonics can be seen clearly. Particularly, the spectrum band of IMF 6 is concentrated around $f_{\mathrm{BPFO}}$, IMF 5 is around $f_{\mathrm{BPFO}}$ and $2f_{\mathrm{BPFO}}$, IMF 4 is around $2f_{\mathrm{BPFO}}$ and $3f_{\mathrm{BPFO}}$, and IMF 3 is around $3f_{\mathrm{BPFO}}$ and $4f_{\mathrm{BPFO}}$. These peaks of the fault frequency harmonics illustrate the tested bearing with outer race defects.

The time length of the signal is *L* = 19 s, and the original signal is of $f_{S}\times L=486,400$ samples. CEEMDAN was used to decompose the original signal directly, and the algorithm was still running after 24 h of operation (the computer processor is I5 2.5 g dual-core, and the operating memory is 8G). The compressed signal is of $f_{\mathrm{ES}}\times L=24,320$ samples, which equals the original signal of 0.95 s. Performing CEEMDAN to decompose the compressed signal 10 times, the mean operation time is 359.2 s. 

For a segment of the original signal, which is of 0.95 s, it consists of the same 24,320 samples as the compressed signal does. Performing CEEMDAN to decompose the signal segment also yields 14 IMFs. The spectra of IMF 7~IMF 12 are shown in Figure 9. It can be seen that the spectrum band of IMF 10 is concentrated around $f_{\mathrm{BPFO}}$, IMF 9 is around $2f_{\mathrm{BPFO}}$, and IMF 8 is around $3f_{\mathrm{BPFO}}$ and $4f_{\mathrm{BPFO}}$. However, none of these harmonics can be seen from these spectra. The reason is that the signal segment to be decomposed is too short, and the times that these harmonics are averaged during FFT are not enough to reduce background noises.

Comparing Figure 8 and Figure 9, it can be seen that PAA enables CEEMDAN to decompose long signals and to yield enhanced diagnostic results.

### 3.2. Validation for Inner Race Defect Case

The vibration signal of a bearing with an inner race defect is shown in Figure 10a. The signal length is L=19 s, the shaft speed is $f_{r}=19.7$ Hz, and the characteristic frequency of the inner race fault is $f_{\mathrm{BPFI}}=97.48$ Hz. Analysing the signal with fast Kurtogram yields the result shown in Figure 11. The diagram is different from the one in Figure 7. The same band is selected, with a center frequency of 10,667 Hz, and a bandwidth of 4267 Hz.

Figure 10b shows the filtered signal for the filtering band, Figure 10c shows the envelope of the filtered signal, and Figure 10d shows the compressed signal of the envelope obtained with PAA. The window length of PAA is also set to be w=20, which satisfies $w\leq 404.20/f_{r}$. The equivalent sampling frequency of the compressed signal is also $f_{\mathrm{ES}}=1.28$ kHz. Figure 10e,f shows the partial enlarged envelope and the partial enlarged compressed signal, respectively. It can be seen that the compressed signal keeps the waveform of low frequency components in the envelope.

Decomposing the compressed signal yields 16 IMFs. The spectra of IMF 3~IMF 6 are shown in Figure 12. It can be seen that the spectrum band of IMF 4 is concentrated around the inner race fault frequency $f_{\mathrm{BPFI}}$. The harmonic of $f_{\mathrm{BPFI}}$ and its sidebands of $f_{\mathrm{BPFI}}\pm f_{r}$, and $f_{\mathrm{BPFI}}+2f_{r}$ are clearly presented in the spectrum of IMF 4. The reason for the modulation frequency of $f_{r}$ is that the inner race defect passes the bearing load zone once every rotation of the shaft, and the transient amplitudes change periodically.

The spectrum of IMF 5 is concentrated around the frequency of $f_{\mathrm{BPFI}}-f_{r}$, and clearly shows the harmonics of $f_{\mathrm{BPFI}}$, $f_{\mathrm{BPFI}}\pm f_{r}$, and $f_{\mathrm{BPFI}}-2f_{r}$. The spectrum of IMF 3 is concentrated around the band of $\left[f_{\mathrm{BPFI}},2f_{\mathrm{BPFI}}\right]$. The harmonic of the fault frequency $f_{\mathrm{BPFI}}$ and its sidebands of $f_{\mathrm{BPFI}}+f_{r}$ and $f_{\mathrm{BPFI}}+2f_{r}$, as well as the second order fault frequency of $2f_{\mathrm{BPFI}}$ and its sidebands of $2f_{\mathrm{BPFI}}-3f_{r}$ and $2f_{\mathrm{BPFI}}-2f_{r}$, can be clearly seen from the spectrum. The sideband of $2f_{\mathrm{BPFI}}-3f_{r}$ can also be seen in the spectrum of IMF 4.

It is worth noting that the characteristic frequency of $f_{\mathrm{BPFI}}=4.948f_{r}$ is very close to the 5th order harmonic of shaft frequency $5f_{r}$. Thus, the fault frequency harmonics and their sidebands are very close to the high order shaft frequencies. In any case, the components of $f_{\mathrm{BPFI}}$, $2f_{\mathrm{BPFI}}$, and their sidebands illustrate that the tested bearing has inner race defects.

## 4. Conclusions

In this paper, a rolling bearing fault diagnosis method that combines PAA and CEEMDAN is proposed. The method firstly extracts the envelope signal from an original signal using bandpass filtering and demodulation, then compresses the envelope with PAA, decomposes the compressed signal with CEEMDAN, and finally investigates the spectra of IMFs. Validation results with real bearings show that the proposed method is effective and efficient.

The interesting components in the original signal for fault diagnosis are the modulation between the fault frequencies and the resonance natural frequencies. The natural frequencies are as high as thousands of Hz, or even tens of thousands of Hz, while the interesting components in the envelope are the fault frequency harmonics demodulated from the original signal. As the fault frequencies are far lower than the natural frequencies, compressing the envelope instead of the original signal avoids information loss.

The spectra of IMFs reflect the average energy over the entire test period. The longer the signal is, the more times the spectra are averaged during FFT, and the better the background noise is reduced. However, in each iteration of CEEMDAN, an IMF is yielded by decomposing tens of signals added with assisted noises, as well as the assisted noises themselves. Thus, CEEMDAN has large memory requirements and low computational efficiency for long signals. Compressing the envelope with PAA enables the use of CEEMDAN for long signals to achieve enhanced diagnosis.

## Figures and Tables

**Figure 1 sensors-22-06599-f001:**
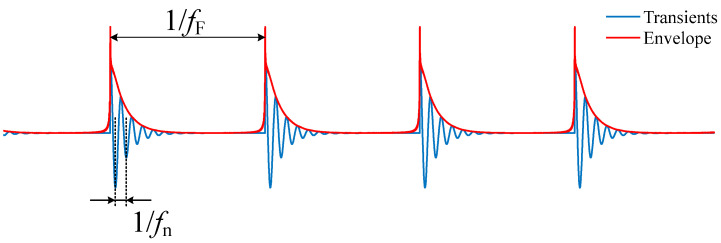
Transient response of bearing defects.

**Figure 2 sensors-22-06599-f002:**
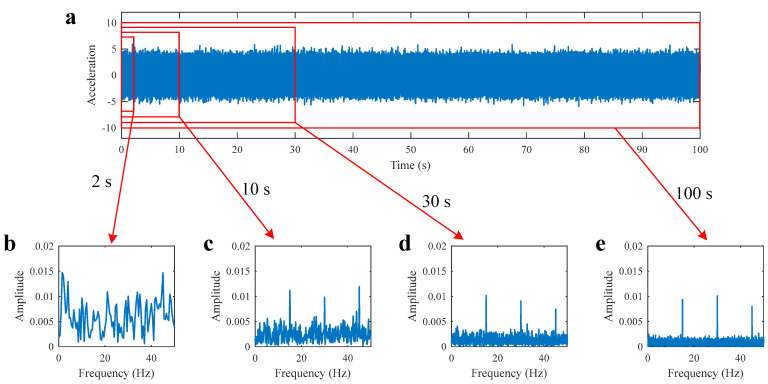
Fault bearing simulation signal and its envelope spectra: (**a**) Time domain signal with noise; (**b**) envelope spectrum of 2 s signal; (**c**) envelope spectrum of 10 s signal; (**d**) envelope spectrum of 30 s signal; (**e**) envelope spectrum of 100 s signal.

**Figure 3 sensors-22-06599-f003:**
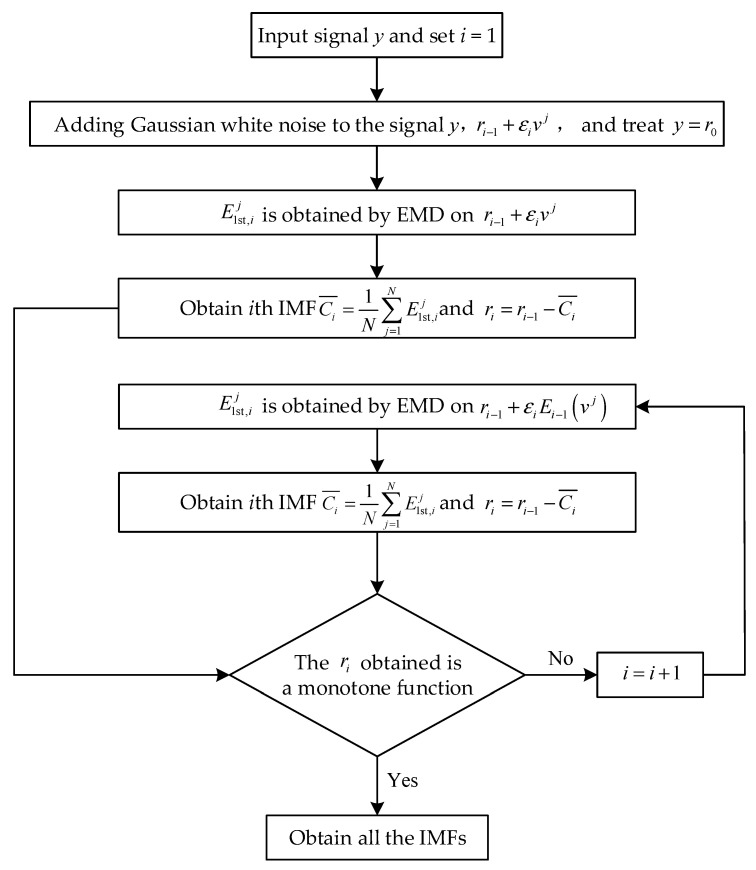
Flow chart of the CEEMDAN algorithm.

**Figure 4 sensors-22-06599-f004:**
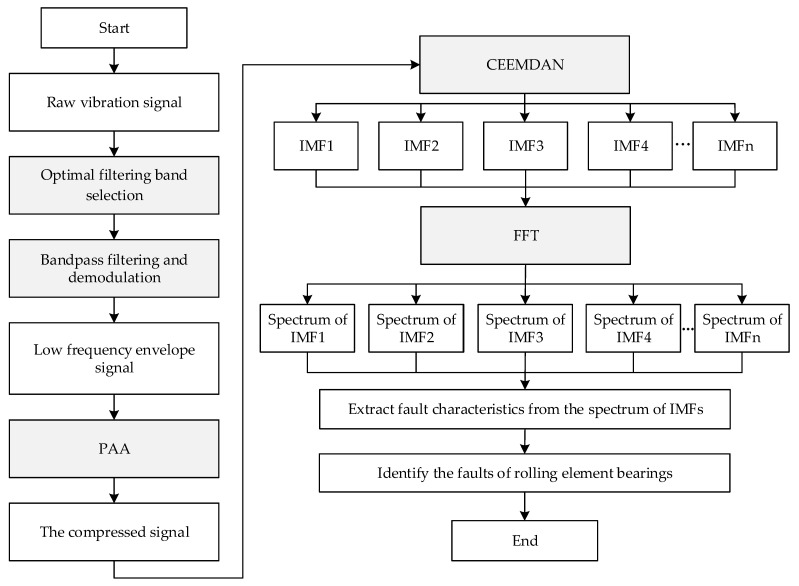
Flow chart of the proposed method.

**Figure 5 sensors-22-06599-f005:**
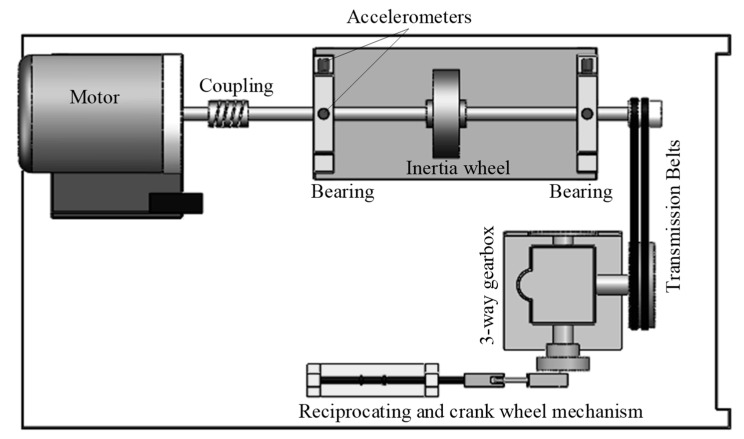
Machinery fault simulation bench.

**Figure 6 sensors-22-06599-f006:**
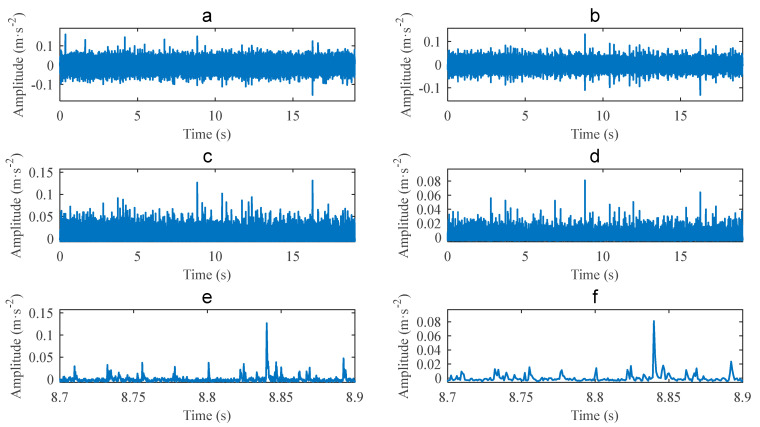
Case 1 for outer race defect: (**a**) Original signal; (**b**) filtered signal; (**c**) envelope; (**d**) compressed signal; (**e**) partial enlarged envelope; (**f**) partial enlarged compressed signal.

**Figure 7 sensors-22-06599-f007:**
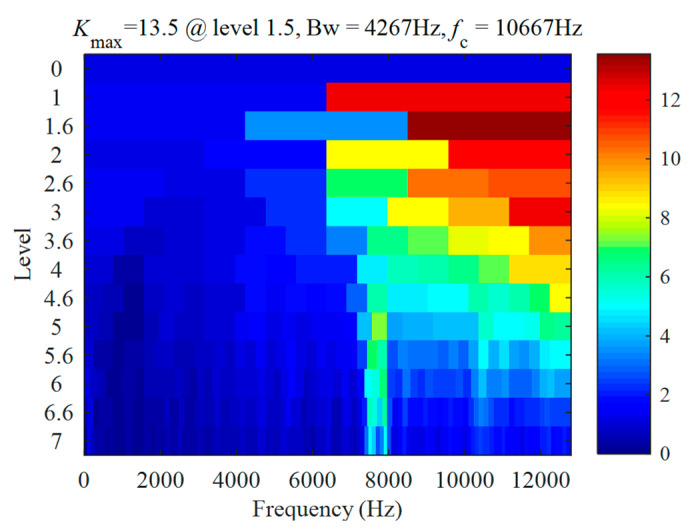
Case 1 for outer race defect: Kurtogram results of the vibration signal.

**Figure 8 sensors-22-06599-f008:**
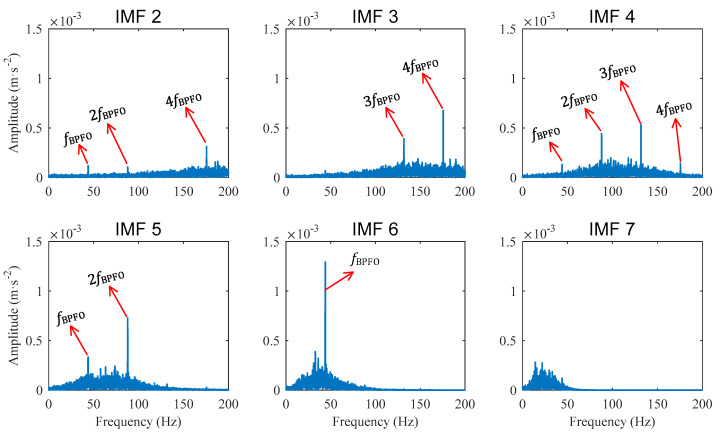
Case 1 for outer race defect: Amplitude spectra of IMFs obtained from compressed signal.

**Figure 9 sensors-22-06599-f009:**
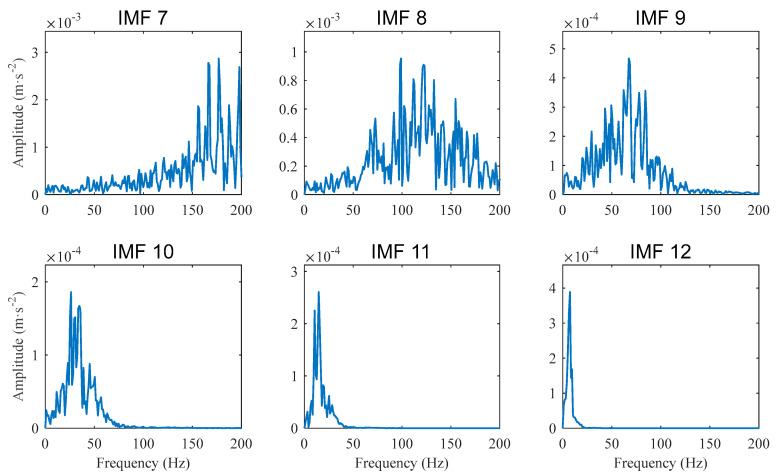
Case 1 for outer race defect: Amplitude spectra of IMFs obtained from an original signal segment that has the same samples as the compressed signal.

**Figure 10 sensors-22-06599-f010:**
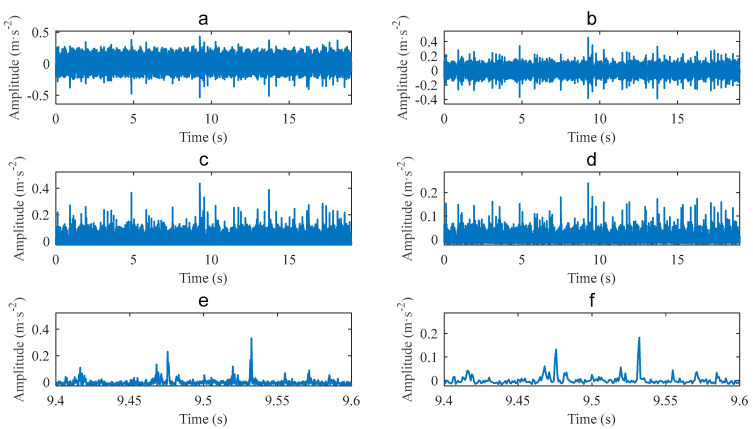
Case 2 for inner race defect: (**a**) Original signal; (**b**) filtered signal; (**c**) envelope; (**d**) compressed signal; (**e**) partial enlarged envelope; (**f**) partial enlarged compressed signal.

**Figure 11 sensors-22-06599-f011:**
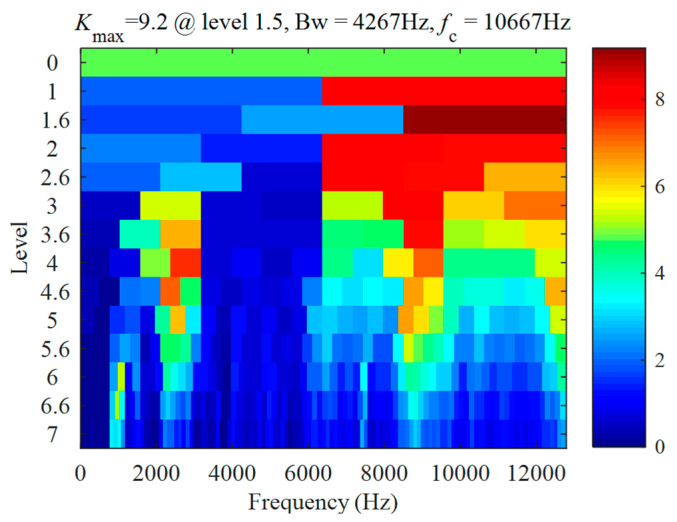
Case 2 for inner race defect: Kurtogram results of the vibration signal.

**Figure 12 sensors-22-06599-f012:**
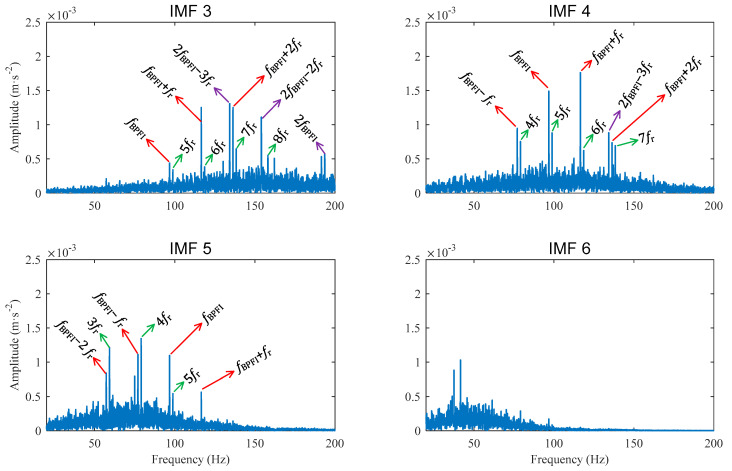
Case 2 for inner race defect: Amplitude spectra of IMFs obtained from the compressed signal.

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
