# Peer review of "Bearing Fault Diagnosis Using Piecewise Aggregate Approximation and Complete Ensemble Empirical Mode Decomposition with Adaptive Noise"

_sensors, 2022, doi:10.3390/s22176599_

Round 1

Reviewer 1 Report

(1) In the proposed algorithm, the first step is to separate low frequency signals from original signals. How to determine the upper limit of filtering which does not lose fault signal components? If so, it may be replaced by reducing sampling frequency.

(2) The piecewise aggregate approximation may be suitable for invariable and persistent fault signals. It may weaken varying signals and be equal to non-piecewise long approximation linearly in results. It is for the case of hardware limit.

(3) The fault signal features exposed depend on SNR value, invariably persistent time, relation with operation frequency, etc. These may be discussed in the algorithm and results. In particular, the difference in the propose algorithm needs to be discussed in results.

(4) As the algorithm is proposed for bearing fault diagnosis, the peculiarity needs to be incorporated in the algorithm. In the last part of page 7, the procedure relevant to signal compression was not explained. Based on the above, the improvement and advantages of the proposed algorithm need to be considered further.

(5) Some expressions may be revised. For example, for “the problem that CEEMDAN cannot handle long signals” in Conclusions, in fact, it depends on hardware capacity.

Reviewer 2 Report

This paper proposes a rolling bearing fault diagnosis method combining Piecewise aggregate approximation with CEEMDAN to avoid some shortcomings in previous CEEMDAN. The purpose of this paper is appreciated but an illogical issue is that the present method has to be compared with the traditional CEEMDAN before its efficiency can be claimed. My detailed comments are as follows.

In the introduction, you have to indicate the superiority of the present method against previous CEEMDAN through some statistical results, instead of just claiming ‘the test data verification results show that the method is effective’.

A broad literature review is desired to attract readers working on a similar topic from different industrial backgrounds. For instance, the application of EMD in [1-2] is desired to be reviewed.

[1] "Nonlinear sparse mode decomposition and its application in planetary gearbox fault diagnosis." Mechanism and Machine Theory 155 (2021): 104082.

[2] "Contact wire irregularity stochastics and effect on high-speed railway pantograph–catenary interactions." IEEE Transactions on Instrumentation and Measurement 69.10 (2020): 8196-8206.

Fig 2 should be optimised. Even the front sizes are not identical in the flow chart.

The benefit of implementing the compression to the signal should be highlighted. Why is the PAA selected instead of other methods?

The most important part is the comparison with existing methods when you present a new diagnosis method. It looks like this essential part is missing in Section 3. Please compensate or discuss.

Round 2

Reviewer 1 Report

It is difficult to determine if the revising without using the Track Changes.

Reviewer 2 Report

I do not have further comments on this paper. Thanks for elaborate revisions.